# Conserved Imprinted Genes between Intra-Subspecies and Inter-Subspecies Are Involved in Energy Metabolism and Seed Development in Rice

**DOI:** 10.3390/ijms21249618

**Published:** 2020-12-17

**Authors:** Lin Yang, Feng Xing, Qin He, Muhammad Tahir ul Qamar, Ling-Ling Chen, Yongzhong Xing

**Affiliations:** 1National Key Laboratory of Crop Genetic Improvement, Huazhong Agricultural University, Wuhan 430070, China; yanglinlin@webmail.hzau.edu.cn (L.Y.); heqin327@163.com (Q.H.); 2College of Life Science, Xinyang Normal University, Xinyang 464000, China; xfengr@163.com; 3State Key Laboratory for Conservation and Utilization of Subtropical Agro-Bioresources, College of Life Science and Technology, Guangxi University, Nanning 530004, China; m.tahirulqamar@gxu.edu.cn

**Keywords:** inter-subspecies, intra-subspecies, conserved imprinted genes, energy pathway, seed development

## Abstract

Genomic imprinting is an epigenetic phenomenon in which a subset of genes express dependent on the origin of their parents. In plants, it is unclear whether imprinted genes are conserved between subspecies in rice. Here we identified imprinted genes from embryo and endosperm 5–7 days after pollination from three pairs of reciprocal hybrids, including inter-subspecies, *japonica* intra-subspecies, and *indica* intra-subspecies reciprocal hybrids. A total of 914 imprinted genes, including 546 in inter-subspecies hybrids, 211 in *japonica* intra-subspecies hybrids, and 286 in *indica* intra-subspecies hybrids. In general, the number of maternally expressed genes (MEGs) is more than paternally expressed genes (PEGs). Moreover, imprinted genes tend to be in mini clusters. The number of shared genes by R9N (reciprocal crosses between 9311 and Nipponbare) and R9Z (reciprocal crosses between 9311 and Zhenshan 97), R9N and RZN (reciprocal crosses between Zhonghua11 and Nipponbare), R9Z and RZN was 72, 46, and 16. These genes frequently involved in energy metabolism and seed development. Five imprinted genes (*Os01g0151700*, *Os07g0103100*, *Os10g0340600*, *Os11g0679700*, and *Os12g0632800*) are commonly detected in all three pairs of reciprocal hybrids and were validated by RT-PCR sequencing. Gene editing of two imprinted genes revealed that both genes conferred grain filling. Moreover, 15 and 27 imprinted genes with diverse functions in rice were shared with Arabidopsis and maize, respectively. This study provided valuable resources for identification of imprinting genes in rice or even in cereals.

## 1. Introduction

Genomic imprinting, also known as genetic imprinting, is an epigenetic phenomenon in which a subset of genes is expressed dependent on the origin of their parents [1]. Genomic imprinting was firstly discovered by Kermicle in maize in 1970, which proved that R pigmentation gene prefers to express maternal allele [2]. Since that, genomic imprinting is widely found in insects, mammals, and flowering plants. Genomic imprinting existed in embryo and placenta in animals, but most imprinting genes were found in the endosperm of flowering plants [3]. Endosperm is one of the products of plant double fertilization, and provided resources for the development of embryo, which contains two maternal genomes and one paternal genome. The expression of imprinted genes is deviated from 2:1. Some of them prefer to express maternal alleles called maternally expressed genes (MEGs). The others prefer to express paternal alleles, called paternally expressed genes (PEGs).

Transcriptomic analysis of developing hybrids is an effective approach to detect imprinted genes in the flowering plants [4]. Several hundreds of imprinted genes have been detected in Arabidopsis, maize and rice through this method [5,6,7,8]. In rice, three different groups used different reciprocal crosses between *Indica* and *Japonica* to detect imprinted genes. The 168 imprinted genes were discovered using the reciprocal cross between 9311 (*Indica*) and Nipponbare (*Japonica*) [7]. About 40% of them was repeatedly detected by Yuan et al. [9]. Moreover, 257 imprinted genes were found using the reciprocal crosses between Longtefu (*Indica*) and 02428 (*Japonica*). In another study, 424 imprinted genes were detected from two different inter-subspecies reciprocal crosses from the *japonica* Liuqianxin and the *indica* Rongfeng, and from the *indica* Wufeng and *japonica* Yu6 [10]. From these studies, 589 imprinted genes were identified in these four crosses. Moreover, 46% of them was commonly detected in at least two sets of four crosses.

In maize, 179 and 100 imprinted genes were detected from the same hybrid between Mo17 and B73, respectively, by two groups [8,11]. Among them, only 56 genes were common. A total of 290 imprinted genes were identified from different development stages of kennels in the hybrid between Mo17 and B73 [12]. Moreover, 81 of them were also detected in one of these two studies. In another study, 177 imprinted genes were identified through five reciprocal crosses among four maize inbred lines [13]; 88% of them were overlapped in at least two crosses.

In Arabidopsis, several groups detected hundreds of imprinted genes through analyzing the transcriptome of endosperm. For example, 208 and 123 imprinted genes were detected from the same reciprocal crosses between Col and Ler by two groups, respectively [5,6]. Only 20 were commonly identified [5]. Later, 66 imprinted genes were identified in the cross between Col and Bur by the third group [14]. Only three imprinted genes were commonly detected in all three studies. In 2019, the transcriptome of endosperm was obtained from two reciprocal crosses (Col/Ler and Col/Tsu) by sequence capture. About 90% of imprinted genes was overlapped between these two crosses when the analysis method for mining imprinted genes was improved [15]. To estimate the conservation of imprinted genes, these imprinting genes identified in both crosses were used to compare with previously reported ones. Only one imprinted gene is commonly detected in all reciprocal crosses mentioned above.

There are several hypotheses to explain the evolution of imprinted genes [16]. Among them, the theory of parent–offspring conflict known as kinship theory suggests that the genomic imprinting is driven by fighting for the allocation of resources from mother to offspring between paternal allele and maternal allele in the endosperm. This theory has been supported by many evidences [17,18,19,20]. It is inferred that the same locus would be selected between two closely related species because of conflict of parents. Moreover, 72% (38/53) of imprinted genes was conserved between *Arabidopsis lyrata* and *Arabidopsis thaliana* that were diverged about 13 Myr ago [19]. In rice, 13 out of 14 imprinted genes selected from cultivated rice were also imprinting in reciprocal crosses between wild rice and cultivated rice [10]. These studies inferred that regulatory mechanism of imprinting was conserved between closely related species.

Significant differentiation of genome constitution and reproductive isolation have been demonstrated between *indica* and *japonica* subspecies [21,22]. If imprinted genes cannot be introgressed between groups, they may evolve different genomic imprinting patterns between subspecies. Therefore, two questions are raised: whether imprinting genes are conserved between *indica* and *japonica* subspecies, or whether more imprinting genes would be detected in inter-subspecies hybrid between *indica* and *japonica* rice than that in intra-subspecies hybrids. To address both questions, we identified the imprinted genes by analyzing the transcriptome data of embryo and endosperm in three pairs of reciprocal crosses including two intra-subspecies crosses and one inter-subspecies cross between *indica* and *japonica*. Consequently, 914 imprinted genes including 801 MEGs and 113 PEGs identified in at least one pair of reciprocal crosses. The imprinted genes tend to be distributed in mini clusters in genomics, and limited imprinting genes were observed between subspecies. Moreover, 15 and 27 imprinted genes are shared between rice and *Arabidopsis thaliana* and between rice and maize, respectively.

## 2. Results

### 2.1. The Number of Genes Expressed in Embryo and Endosperm

We used the criterion of FPKM (fragments per kilobase of exon model per million) more than 0.5 to define the expressed gene in the embryo and endosperm [23]. It was interesting that the number of expressed genes in the embryo is 5% more than that in the endosperm in all crosses, except the crosses of N9 and NZ with equivalent number of expressed genes in both organs (Table 1). In 9N, the largest number of more than 24,000 genes were expressed in embryo while less than 23,000 genes were expressed in endosperm. The least number of genes were expressed in the *indica* intra-subspecies hybrid reciprocal crosses between 9311 and Zhenshan97 (R9Z) in both organs, 20% less than the other two sets of reciprocal hybrids (Table 1). However, an equivalent number of genes were expressed in embryo and endosperm in all sets of reciprocal crosses.

### 2.2. The Number of Parental Polymorphic Genes Expressed in Embryo and Endosperm of Hybrids

RNA was extracted for sequencing from embryo and endosperm 5–7 days after pollination (DAP) from three sets of reciprocal hybrids. All hybrids were identified as real hybrids by InDel markers (Appendix A). Moreover, 5–9 million single-end reads were generated for hybrids of 9N in both embryo and endosperm (Appendix A). About 20 million paired-end reads were sequenced for hybrids of N9, 9Z, Z9, ZN, and NZ in both embryo and endosperm (Appendix A). The number of Single Nucleotide Polymorphisms (SNPs) from 56,332 to 202,828 were detected in inter-subspecies hybrid between 9311 and Nip (Table 1). 29,136 to 52,875 SNPs were found in the *indica* intra-subspecies hybrid between 9311 and ZS97 (Table 1). The SNPs from 8842 to 14,735 in the *japonica* intra-subspecies hybrid between ZH11 and Nip were detected (Table 1). This observation suggested that the polymorphism between *indica* and *japonica* parents was higher than that of intra-subspecies parents. There are 24,598 expressed genes in embryo and 24,558 genes in endosperm with SNPs between parents in reciprocal crosses between 9311 and Nipponbare (R9N). The corresponding numbers of genes with SNPs between parents were decreased to 8330 and 7180 in R9Z (Table 1). Moreover, the number of such genes were further decreased to 2891 and 2462 in RZN.

InDels were also identified between each pair of parents. Similar to SNPs, the largest number of expressed genes of 12,499 in embryo and 10,864 in endosperm exhibited insertion/deletion polymorphism in R9N (Table 1). There were 1795 and 986 genes, and 492 and 307 genes showing InDels in embryo and endosperm in R9Z and RZN, respectively. These expressed genes showing either polymorphism of SNPs or InDels were used for analysis of origin of parents.

### 2.3. Identification of Imprinted Gene from Three Sets of Reciprocal Crosses

Most genes in the three combinations exhibited a maternal to paternal ratio of 2:1 in endosperms (Appendix A), and of 1:1 in embryos (Appendix A). This was consistent with the expected parental genomic contribution in triploid endosperm and in diploid embryo, indicated that our dataset is suitable to identify imprinted genes.

In general, more than 200 imprinted genes for each pair of reciprocal crosses were found in the endosperm, less than 10 imprinted genes were found in the embryo (Table 2). In R9N, 546 genes were parentally biased expression in the endosperm, including 482 MEGs and 64 PEGs, no parentally biased gene was observed in the embryo (Table 2, Appendix A). In R9Z, there were 286 parentally biased genes in the endosperm, of which, 235 and 52 genes showed maternally and paternally preferred expression in the endosperm, respectively. One gene was paternally preferred expression in the embryo (Table 2, Appendix A). In RZN, 211 parentally biased genes, including 195 MEGs and 16 PEGs, were identified in the endosperm. Three parentally biased genes were identified in the embryo; they all showed maternally preferred expression (Table 2, Appendix A).

### 2.4. Preference Expression of Imprinted Genes in Endosperm

Opposite to the case of total expressed gene numbers, the number of imprinted genes in the endosperm were far more than that in embryo, and the expression level of imprinted genes in the endosperm were higher than that in the embryo. In R9Z, the expression of MEGs was higher in the endosperm than in the embryo, but this phenomenon was not observed for PEGs (Appendix A). In R9N, expression level of both MEGs and PEGs identified in endosperm was two-fold higher in the endosperm than that in embryo (Appendix A). The imprinted genes detected in RZN exhibited the similar expression pattern of higher expression in endosperm than in the embryo (Appendix A).

### 2.5. Confirmation of Maternally Expressed Genes

Five MEGs (*Os01g0151700*, *Os07g0103100*, *Os10g0340600*, *Os11g0679700*, and *Os12g0632800*) were detected in all three sets of reciprocal crosses. These imprinted genes were further validated by sequencing RT-PCR products. All five MEGs were only expressed the female parental alleles in all three pairs of reciprocal crosses (Figure 1A–E). The ratio of the expression amounts of maternal alleles to that of paternal alleles accounted for more than 4:1 in endosperm (Figure 1F). Six randomly selected MEGs commonly detected in two pairs of reciprocal crosses were also validated by sequencing (Appendix A). For example, *Os02g0626400* and *Os11g0455500* were maternally expressed in R9N and R9Z. These results demonstrate that the analysis method in this study was reliable.

### 2.6. Mini Cluster of Imprinted Genes

If two or more imprinted genes are tandemly located in physical position without an intergenic region in the genome, we called them mini cluster of imprinted genes. Moreover, 146 imprinted genes in R9N comprised 72 mini clusters, including seven paternal-only (the cluster consists of PEGs only) and 65 maternal-only (the cluster consists of MEGs only) clusters (Appendix A). Among them, two clusters each contained three imprinted genes (Appendix A). While in R9Z, there were 15 clusters including one paternal-only cluster, two maternal-paternal clusters (the cluster contained PEGs and MEGs), and 12 maternal-only clusters (Appendix A). Each of them contained two imprinted genes. In RZN, 82 imprinted genes formed 38 mini clusters, including two paternal-only clusters, one maternal–paternal cluster, and 35 maternal-only clusters. Among them, 34 were two-gene mini clusters, two were three-gene mini clusters, and two were four-gene mini clusters (Appendix A). Shortly, 26.7%, 10.5%, and 38.9% of imprinted genes were in cluster in R9N, R9Z, and RZN, respectively.

To test the reliability of cluster genes, we simulated the expectation possibility that two randomly selected genes are neighbors. We randomly extracted 200, 500, and 1000 genes from the genome, calculated the number of adjacent neighbor genes, and repeated 10,000 times. The simulation results showed that 95% confidence interval of the number of randomly selected genes in neighbor pairs was lower than 5% of the all selected genes (Figure 2). In all paired reciprocal crosses, the percentage of clustered imprinted genes was much higher than 5% of imprinted genes. That was, the proportion of imprinted genes in neighbor pairs was significantly higher than the simulated value. Therefore, the imprinted genes tended to form mini clusters.

### 2.7. Conservation of Imprinted Genes between Subspecies in Rice

In this study, 914 imprinted genes were detected. Among them, 72 imprinted genes were shared between R9N and R9Z, 16 and 46 imprinted genes were commonly detected between RZN and R9Z, and between R9N and RZN, respectively. Almost 80% of commonly identified imprinted genes between any two pairs of reciprocal crosses was MEG genes. There were five MEGs commonly identified in all these three pairs of reciprocal crosses (Figure 3). We analyzed the reason why the overlapped imprinted genes were limited between/among these three combinations (Table 3). The total number of imprinted genes was 913 among three combinations. Among them, 546 genes were imprinted in R9N, 260 genes were not significantly deviated from the expected ratio of 2m:1p by Chi-square test. However, 48 genes could not identify because there was no SNP between parents. The remaining 60 genes were not detected because there were no reads across the SNP. In other two combinations, most of the imprinted genes were not detected because of there were no SNP between parents.

In order to specify whether imprinted genes were conservative, a random simulation was conducted based on the number of imprinted genes in the three pairs of reciprocal crosses. In the simulation, 37,852 genes annotated by RAP-DB (Rice Annotation Project Database) were used as background; 546, 286, and 211 genes with the same number of imprinted genes in the three pairs of hybrid combinations were extracted at a time and simulated 10,000 times randomly. The simulation results showed that 95% confidence interval of the number of overlapping genes in the three pairs of combinations was 0 (Appendix A). Therefore, five imprinted genes shared by the three pairs of reciprocal crosses was much higher than the random simulation values, indicated the imprinted genes were conservative (Appendix A). In addition, we also simulated the random number of common genes shared by two pairs of reciprocal crosses. The simulation results showed that the expectation number of shared genes with 95% confidence intervals was 8, 6, and 4 between R9N and R9Z (Appendix A), between R9N and RZN (Appendix A), and between R9Z and RZN (Appendix A), respectively. In fact, the number of shared imprinted genes were 72, 46, and 16, respectively, which were far beyond the expected numbers in random simulation experiments.

A total of 546 imprinted genes had been previously detected in rice [10]. Moreover, 154 imprinted genes of them were shared with this study. In case of the number caused by chance, a simulated experiment was conducted. In the simulation, 546 genes and 913 genes were randomly picked up from the RAP-DB each time, and the number of overlapped genes was recorded. After simulation of 10,000 times, the number of overlapped genes was 21 in the 95% confidence interval. Thus, the number of detected conserved imprinted genes were much more than 21 between this study and other studies, indicating conserved imprinted genes are reliable (Appendix A).

### 2.8. Two Imprinting Genes Verified for Grain Filling

Five imprinted genes were detected across three combinations. They have different functions (Table 4). To identify the function of imprinted genes, two (*MEG1* and *MEG2*) of five commonly identified MEGs across the three pairs of reciprocal crosses were edited in ZH11. *MEG1* (*Os12g0632800*) encodes a receptor-like protein kinase. Its homologous gene *HAIKU2* in Arabidopsis regulated seed size [24]. Two mutants (*meg1-1* and *meg1-2*) were generated for *MEG1* (Figure 4A). A 16-bp deletion and an 8-bp deletion were detected in *meg1-1* and *meg1-2*, respectively, which resulted in frame shift mutation (Figure 4A). The grain filling process of both mutants was slower than that of WT (wild type) from 14DAF to seed mature (Figure 5A). Finally, thousand-grain weight of mutants were reduced by 20% (Figure 5C). *MEG2* (*Os10g0340600*) encodes a beta galactosidase (BGAL) involved in cell wall synthesis. The mutation of its homolog in Arabidopsis caused shorter siliques [25]. Two frame-shift mutations (*meg2-1* and *meg2-2*) were generated (Figure 4B). Like *MEG1*, the mutants had a slower grain filling speed and a reduced thousand-seed weight by 10% as compared with wild type (Figure 5B,D).

### 2.9. MEGs Involved in Carbohydrate Metabolism while PEGs Involved in Signal Transduction

To infer the functional MEGs and PEGs, we gathered all the MEGs and PEGs detected in three pairs of reciprocal crosses for gene ontology enrichment analysis, respectively. We presented the results in GO (Gene Ontology) terms of biological process, molecular function, and cellular component.

Most of the MEGs were enriched in carbohydrate metabolism and biosynthetic process, nutriment metabolic of nitrogen and transmembrane transport; and some MEGs were enriched in molecular functions, such as hydrolase activity, cellulose synthase activity, and transferase activity; and very few MEGs were involved in cell membrane (Appendix A). Whereas PEGs were only enriched in molecular functions including protein serine/threonine kinase activity, protein kinase activity, kinase activity, and phosphotransferase activity (Appendix A). These results indicated that MEGs prefer to involve in the metabolism, biosynthesis, and transport of nutriment, and PEGs involve in signal transduction.

### 2.10. Subcellular Location of Proteins Encoded by Imprinted Genes

The subcellular location of proteins encoded by the MEGs (PMEGs) and PEGs (PPEGs) were analyzed in the website of RiceNetDB (http://bis.zju.edu.cn/ricenetdb/search.php). The 705 PMEGs and 106 PPEGs were separated into various subcellular locations. Moreover, 40% of the PPEGs were in nucleus (Figure 6, Appendix A). Unlike PPEGs, PMEGs were in all organelles (Figure 6). More than 215 PMEGs (30%) were in chloroplast (Figure 6, Appendix A). The plastid of endosperm is functionally similar to chloroplast, the organelle of starch synthesis. This result indicated that the MEGs played important role in starch synthesis.

### 2.11. KEGG (Kyoto Encyclopedia of Genes and Genomes) Pathway Enrichment Analysis

PEGs were only enriched in phosphatidylinositol signaling system, which is related to gene regulation (Appendix A). MEGs were enriched in many metabolic pathways (Appendix A). Most of them involves in carbon metabolism, and some of them were enriched in biosynthesis of secondary metabolites such as flavonoid, carotenoid, stilbenoid, diarylheptanoid, and gingerol, isoquinoline alkaloid.

The starch and sucrose metabolisms were focused because seed is the storage organ of nutrition and energy. Eight MEGs (*Os01g0605700*, *Os01g0606000*, *Os02g0534400*, *Os03g0736300*, *Os06g0247900*, *Os11g0236100*, *Os06g0256500*, and *Os01g0749400*) were involved in carbohydrate metabolic pathway (Figure 7). Among them, *Os02g0534400* and *Os03g0736300*, known as *OsCIN1* and *OsGH9A3*, respectively, were characterized. *OsCIN1* is a cell wall invertase that play important roles in grain filling of rice [26]. *OsGH9A3* was involved in cellulose biosynthesis [27]. Seven MEGs took part in carbon fixation, which supplied intermediate products to other pathways (Appendix A). Of them, *Os12g0291100* (*OsRBCS3*) and *Os12g0292400* (*OsRBCS4*) encode rubisco small subunits. *Os01g0723400* (NADP-ME2) participates in photosynthesis. Eighteen MEGs were involved in the phenylpropanoid biosynthesis pathway (Appendix A). This pathway plays an important role in secondary metabolites, resistance of plant, aromatic compound, and aromatic amino acid biosynthesis. For example, *Os02g0627100*, known as *OsPAL4,* encodes phenylalanine ammonia lyase, which is related to disease resistance. *Os02g0187800* (*OsCAD2*) participates in the synthesis of lignin of rice internode.

Sixteen MEGs were involved in the plant hormone, including auxin, ABA, ethylene, and JA signal transduction (Appendix A). *Os01g0190300*, known as *OsIAA2,* is a member of the Aux/IAA auxin responsive gene family. *Os05g0213500* known as *OsPYL5* is the receptor of ABA. Auxin and ABA take part in seed development. These MEGs may regulate seed development through the signal transduction of auxin and ABA.

Four PEGs were enriched in the phosphatidylinositol signaling system. Three of them belonged to inositol 1,3,4-trisphosphate 5/6-kinase (ITPK). Mutations of *OsITPK6* effectively reduced phytic acid biosynthesis in rice grain [28].

## 3. Discussion

The imprinted genes were moderate conservation across subspecies in rice. The number of genes expressed in endosperm with SNPs between parents was 24,558, 7180, and 2462 for R9N, R9Z, and RZN crosses, respectively (Table 1). The number of imprinted genes was 546, 211, and 286 for the three pairs of crosses, respectively. Although the polymorphism between parents ZH11 and Nip was the least, the proportion of detected imprinted genes was the highest (16%) in RZN. In contrast, the proportion was about 5% in both R9N and R9Z. Thus, the number of imprinted genes was not correlated with the number of polymorphic genes between parents.

In this study, hundreds of imprinted genes were identified in each reciprocal cross. Moreover, 9%, 6%, and 3% imprinted genes were commonly detected between R9N and R9Z, between R9N and RZN and between R9Z and RZN, respectively. Only five imprinted genes were overlapped among three pairs of reciprocal crosses (Figure 3). Although the number of overlapped imprinted genes was low between/among these three pairs of reciprocal crosses, they were much higher than the expected ones by chance. Therefore, gene imprinting is conserved between subspecies, but only a small partial of imprinted genes is conserved. Moreover, these five commonly detected imprinting genes have potential diverse functions, such as regulating seed size, regulating the importing protein of chloroplast, involving cell wall synthesis, and anther development (Table 4).

Imprinted genes are frequently involved in energy metabolism and seed development. The theory of parent-offspring conflict has more supporting evidence than the other hypotheses to explain the evolvement of imprinted genes [29]. This hypothesis suggests that the function of the imprinted genes should be enriched in the synthesis and transport of nutrients. In mammals, most imprinted genes are indeed involved in the synthesis and transport of nutrients [30,31]. In this study, we performed transcriptome analysis for three pairs of reciprocal crosses that represent intra-subspecies crosses of *Indica* (R9Z), *Japonica* (RZN), and inter-subspecies crosses between *Indica* and *Japonica* (R9N). It is also found that MEGs were enriched in carbohydrate metabolism and biosynthetic process, nutriment metabolic of nitrogen, and transmembrane transport in rice (Appendix A). Among 102 MEGs involved in metabolic pathways, 13, 13, and 9 were related to starch and sucrose metabolism, biosynthesis of amino acids, amino sugar, and nucleotide sugar metabolism. Imprinting in angiosperms is predicted to regulate seed development [29], but the evidence is very limited [32]. For example, MEGs, such as *OsCIN1* and *OsGH9A3,* play important roles in rice seed development [26,27]. In this study, *Os12g0632800* is a MEG. Its homolog in Arabidopsis regulates seed size [24]. *Os10g0340600* is a MEG. The mutation of its homolog in Arabidopsis caused shorter siliques [25]. Gene editing confirmed that both MEGs regulate grain filling and endosperm development in rice (Figure 4). These results indicate that MEGs involved in energy metabolism and seed development.

In a previous study, seven and 53 imprinted genes were commonly identified between rice and Arabidopsis and between rice and maize, respectively [10]. It was inferred that, the reason why the number of conserved imprinted genes was low between species is that polymorphisms in many genes between parents did not exist in two species. Thus, only a partial of genes could be used for conservation analysis of imprinted genes between species. It is necessary to comprehensively identified imprinted genes through investigating more reciprocal crosses. Compared with previous studies, 15 and 27 imprinted genes detected in this work were commonly detected in Arabidopsis and maize, respectively (Appendix A). The common imprinted genes were still limited between species and no imprinted genes were commonly detected in three species. These results indicated that gene imprinting may have been evolved independently among these species.

Among the 15 common imprinted genes between rice and Arabidopsis, seven genes were characterized (Appendix A). Of them, five imprinted genes, including three MEGs (*UMAMIT25*, *SULTR3.1,* and *UGD2*) and two PEGs (*NOT2a* and *VIM1*) were involved in seed development. UMAMIT25 mediated amino acid transport in the seed of Arabidopsis [33]. Sulfur was a very important macronutrient for life. SULTR3.1 belonged to the group 3 plasmalenna-predicted sulfate transporters (SULTR3) that was involved in supplying sulfate to the embryo and was important for the synthesis and maturation of embryo proteins in Arabidopsis [34]. Nucleotide sugars are the main components of plant cell wall that is formed during endosperm cellularization [35]. UDP-glucose dehydrogenase (UGD) played important roles in the synthesis of nucleotide sugars [36]. Thus, UGD2 is likely related to seed development. Both NOT2a and NOT2b are NOT2_3_5 domain containing-proteins. They physically interact with the conserved DCL1 (DICER-LIKE1) protein between rice and Arabidopsis to mediate the biogenesis of miRNAs (MicroRNAs) in Arabidopsis [37]. It is well known that miRNAs is involved in many plant development processes, including seed development [38]. VARIANT IN METHYLATION 1 (VIM1) cooperated with DNA methyltransferase 1 (MET1) to maintain the CpG methylation of the genome in Arabidopsis [39]. Most other imprinted genes are not characterized but are likely involved in plant responses to stress. It is noted that the endosperm of rice is undergoing continuous cell proliferation and nutrient storage in the sampling period, whereas the endosperm begins to break down in Arabidopsis, and the cotyledon becomes the primary storage organs [4]. The conserved imprinted genes between rice and Arabidopsis likely have diverse function in seed development.

Among the 27 common imprinted genes between rice and maize, the functions of nine genes were recorded (Appendix A). Some of them were important in seed development, including three MEGs (*OsMADS13*, *OsNPF2.4,* and *LTP2*) and two PEGs (*OsYUCCA11* and *OsITPK1*). *ZmMADS1a*, the homolog of *OsMADS13* in maize was involved in starch biosynthesis by regulating the expression of starch biosynthesis genes during maize seed development [40]. *OsNPF2.4* is a *NO*_3−_ transporter belonged to the NPF family (*NO*_3−_/peptide transporter; originally named *NRT1/PTR*), which is involved in the long distance transport of *NO*_3−_ in rice [41]. *OsITPK1* involved phytic acid biosynthesis. Phytic acid is the main storage form of phosphorous in cereal seed [28]. LTP2 (lipid transfer protein 2) involved the transport of lipids from the polar to the outer cell of embryo in maize embryogenesis [42]. *OsYUCCA11* regulated accumulation of Auxin in rice seed development [43]. In the sampling period, the endosperm of both rice and maize underwent continuous cell proliferation and nutrient storage. Thus, the conserved imprinted genes would likely involve starch synthesis and energy metabolism.

## 4. Materials and Methods

### 4.1. Plant Materials

Two *indica* cultivars 9311 and Zhenshan 97 (ZS97) and two *japonica* cultivars Nipponbare (Nip) and Zhonghua 11 (ZH11) were used as parents to develop three pairs of reciprocal crosses by artificial emasculation and pollination. All plants were planted in the field at the experimental station of Huazhong Agricultural University (Wuhan, China) in the summer of 2012. To obtain the inter-subspecies reciprocal crosses of R9N which included the cross 9N with 9311 as the female parent and Nip as the male parent, and the cross N9 with Nip as the female parent and 9311 as the male parent, the female parental plants were emasculated and fertilized with the pollens of the male parents after heading. Accordingly, the *japonica* intra-subspecies reciprocal crosses of RZN, including the cross ZN with ZH11 as the female parent and the cross NZ with Nip as the female parent, and the *indica* intra-subspecies reciprocal crosses R9Z, including the cross 9Z with 9311 as the female parent and the cross Z9 with ZS97 as the female parent, were obtained (Table 5).

### 4.2. Identification of Hybrids and RNA Extraction

The hybrids were separately harvested one by one at 5–7 days after pollination. A total of 20–30 hybrids were harvested for each cross. Each seed was dissected into embryo and endosperm. A part of endosperm was used to extract DNA for hybrid identification by DNA markers. DNA extraction followed CTAB (hexadecyl trimethyl ammonium bromide) protocol. Three polymorphic InDel markers RID2-1 (rice InDels marker), RID9-21, and RID6-9 were used for identification of hybrids for the reciprocal crosses of R9N, R9Z, and RZN, respectively [44]. The amplified products were separated by 1.5% agarose gel electrophoresis. The embryos and endosperms of 20–30 identified hybrids from the same cross were cumulated for RNA extraction, separately. TRIzol reagent (Invitrogen, Shanghai, China) was used to extract RNA following the protocol. A total of 12 RNA samples were obtained for these three reciprocal crosses.

### 4.3. Construction of RNA and DNA Sequencing Library

A total of 14 RNA sequencing libraries were constructed (Appendix A). For the three pairs of reciprocal crosses R9N (9N and N9), R9Z (9Z and Z9), and RZN (ZN and NZ); RNA sample was used to construct the library from the embryo and endosperm respectively. These libraries were sequenced to produce 2 × 101 bp paired reads. For N9, two libraries were constructed from the same embryo RNA sample to produce 35-bp single reads and 50-bp single reads. Like the case in embryo, such two libraries were constructed from the same endosperm RNA of N9. All of the RNA sequencing libraries were constructed following the RNA sequencing protocol provided by the Shanghai Majorbio Bio-Pharm Technology Co. Ltd. (Shanghai, China). Briefly, the poly A containing mRNA was enriched and fragmented. Double-stranded cDNA was obtained using reverse transcription kit (Invitrogen, Shanghai). Sequencing adaptor was added to the end of these fragmented cDNAs and then the cDNA libraries were sequenced using Illumina platform.

The sequencing library of the genomic DNA extracted from leaves of ZH11 following the CTAB protocol was constructed following the suggested protocol (Illumina Inc., San Diego, CA, USA). Paired-end sequencing libraries with an insertion of approximately 300 bp were sequenced on an Illumina HiSeq 2000 sequencer at Berry Genomics Company (Beijing, China). The ZH11 DNA sample was sequenced at a length of 101 base pairs by paired-end sequencing (Appendix A).

### 4.4. Construction of Pseudo Genomes of ZH11 and ZS97

As the genome of ZH11 is unavailable, the pseudo-genome of ZH11 was constructed using Nip genome as reference. The DNA of ZH11 was deeply (30×) sequenced. Compared with the reference genome of Nip, SNPs and InDels between ZH11 and Nip were identified. In order to get precise SNPs and InDels information, the duplicated pair-end reads were removed using local perl script before mapping, and then the low-quality base of less than 10 reads was trimmed, and the read length less than 27 nt was filtered using fastx-toolkit (download from website http://hannonlab.cshl.edu/fastx_toolkit/index.html). After that, the reads were mapped to the genome of Nip using the BWA (Burrows-Wheeler Aligner)(bwa6.1) [45]. SNPs and InDels were validated using samtools and Genome Analysis Toolkit [45,46]. Therefore, the Nip genome was substituted with SNPs and InDels between ZH11 and Nip, and the pseudo genome of ZH11 (pgZH11) was developed using our in-house perl script, such as the method used in Arabidopsis [6]. Similarly, the pseudo-genome of ZS97 (pgZS97) was constructed using 9311 genome as reference. The raw genome data of ZS97 came from the public data [47].

### 4.5. Identification of Cultivar Specific Reads

The reads from the endosperm and embryo RNA-seq of different reciprocal crosses were mapped to their own parental genomes including Nip genome, 9311 genome, pgZH11, and pgZS97 separately by Bowtie [48]. The pair-end RNA-seq reads were mapped to the parental genomes independently and then counted together. If a read could be mapped to both genomes of parents with few mismatches, it was assigned to the genome [6]. The read with a clear origin of parent genome is named as allele specific read (ASR). In order to make the analysis easier, the mismatching positions were transferred to the Nip genome positions in R9N and RZN. The Nip genome version is Os-Nip-Reference-IRGSP-1.0 (IRGSP-1.0) with annotation from The Rice Annotation Project (RAP-DB) [49]. The *indica* 9311 genome and annotation were extracted from the Rise database of Beijing Genomics Institute [50].

### 4.6. Identification of Imprinted Genes

The ASR number of each gene was calculated using coverageBed from bedtools [51]. A χ2 test was performed to determine whether the maternal ASRs: paternal ASRs of each gene was significantly deviated from the expected ratio of 2m:1p in endosperm and 1m:1p in embryo respectively [52]. Genes significantly deviated from the expected expression ratio of maternal allele to paternal allele (χ^2^ > 3.84, *p* < 0.05) and with more than 4-fold change m:p in endosperm and/or 2-fold change m:p in embryo in both reciprocal hybrids were identified as MEGs. On the contrary, genes significantly deviated from the expected ratio (χ^2^ > 3.84, *p* < 0.05) and with more than 1-fold change p:m in endosperm, and 2-fold change p:m in embryo in both reciprocal hybrids were identified as PEGs.

### 4.7. Transferring 9311 Gene ID and MSU (Michigan State University) Gene ID to RAP-DB Gene ID

To make convenient comparison of imprinted genes among three pairs of reciprocal hybrids, the imprinted genes identified from R9Z were transferred to RAP-DB ID. Firstly, the 9311 cDNA were aligned to RAP-DB cDNA using BLAT (BLAST-like alignment tool) [53], and then the 9311 gene ID was transferred to the RAP-DB ID with perl script. The MSU gene ID was transferred to the RAP-DB gene ID based on MSU-RAP.txt downloaded from RAP-DB [49].

### 4.8. Validation of Imprinted Genes

Primers for RT-PCR sequencing were designed to amplify gene fragments harboring informative SNPs (Appendix A). About 3 μg total RNA from hybrid endosperm was used for cDNA synthesis using an Invitrogen Superscript III Reverse Transcriptase kit. The 20 μL PCR mix contained 2 μL of 10 × buffer, 1.3 μL of MgCl_2_ (25 mM), 1 μL of dNTPs (5 mM), 0.5 μL of each primer (20 μM), and 1-unit Taq DNA polymerase. The thermal cycler program included an initial incubation of 3 min at 95 °C, followed by 27–35 cycles, depending on the experimentally determined level of gene expression, where each cycle consisted of 20 s at 95 °C, 20 s at 55 °C and 1 min at 72 °C, followed by 2 min at 72 °C and 1 min at 25 °C. PCR products were visualized on a 1.5% agarose gel with ethidium bromide. RT-PCR products generated from endosperm were submitted for direct sequencing using the Sanger method to get the relative expression of the two parent alleles.

### 4.9. Identification of Conserved Imprinted Genes among Species

In order to find conserved imprinted genes among rice, maize, and Arabidopsis, bi-directional BLASTP (basic local alignment search tool for proteins) was used. For comparison between species, the proteins encoded by imprinted genes in species A were aligned to that in species B firstly by BLAST, and then the proteins in species B were aligned to that in species A. As the top score of hit rate cannot reflect the order of rigorous Smith–Waterman scores appropriately, we define bi-directional hit rate (BHR) as: BHR = Rf × Rr [54]. Rf represents the blast score from forward (A against B). Rr represents the score by blast reverse (B against A). To get the best hits, the threshold of BHR score was set 0.95. All paired comparison among rice, maize, and Arabidopsis were made.

### 4.10. Calculation of Abundance of Gene Expression

The reads of R9N and RZN were mapped to Nip genome using tophat2 [55]. The reads of R9Z were mapped to 9311 genome using tophat2 tool. Then the cufflinks were used to calculate expression abundance for every gene in all samples by supplied reference annotation files [56].

### 4.11. Gene Ontology (GO) Analysis, Subcellular Location, and KEGG Pathway Analysis

The website tool agriGO v2 (http://systemsbiology.cau.edu.cn/agriGOv2/) is applied to GO analysis. The imprinted genes detected in this study were uploaded to the website tool RiceNetDB (http://bis.zju.edu.cn/ricenetdb/) to analyze the subcellular location. The KEGG pathway was analyzed through the KEGG mapper [57].

### 4.12. Generation of Knockout Mutants of Two Imprinted Genes

Gene mutation was produced by CRISPR/Cas9 system as described in previous study [58]. The oligos that harbored NGG (PAM) were used in the construction of the sgRNA vectors for the two candidate MEGs, *MEG1* (*Os12g0632800*) and *MEG2* (*Os10g0340600*). Two targets that located in different exons were chosen for each gene. These two target fragments were introduced into two sgRNA expression cassettes by dual-nested PCR, driven by OsU6b and OsU3 promoters. The multiple sgRNA expression cassettes were then ligated to the CRISPR/Cas9 binary vector (pYLCRISPR/Cas9Pubi-H) based on Golden Gate cloning. After that, these vectors were introduced into Agrobacterium tumefaciens strain EHA105, and were separately transferred into ZH11 using Agrobacterium-mediated transformation. The sequence mutations were verified by PCR and sequencing. The primers used for construction of vector and verify the mutation were listed in Appendix A.

### 4.13. Measurement of Grain Filling

The grain filling of homozygous mutant lines (T2 generation) and wild type plants were investigated. The main-stem panicles were tagged when they headed. Five tagged panicles per genotype were gradually harvested starting at 7 days after flowering (DAF) at a 7-day interval until maturity including 7 DAF, 14 DAF, 21 DAF, 28 DAF, and 35 DAF. The filling grains of panicle were harvested and oven-dried at 70 °C for about 72 h to a constant weight. The grains of the whole panicle were weighted and counted. The hundred-grain weight (HGW) or thousand-grain weight (TGW) was calculated as follows: HGW = (grain weight/the number of grains) × 100. TGW = (grain weight/the number of grain) × 1000.

## Figures and Tables

**Figure 1 ijms-21-09618-f001:**
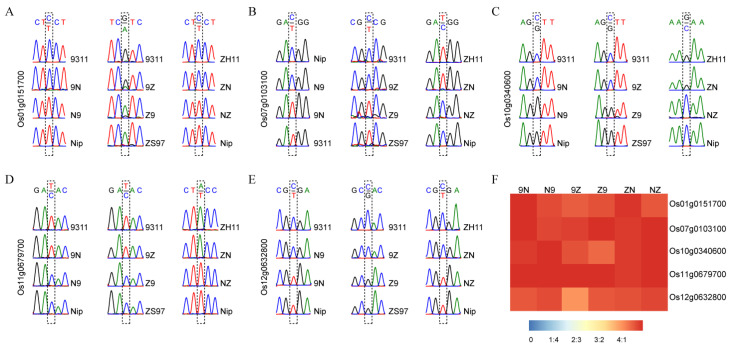
Validation of five conversed imprinted genes in endosperm. (**A**–**E**) The RT-PCR sequencing results of *Os01g0151700*, *Os07g0103100*, *Os10g0340600*, *Os11g0679700*, and *Os12g0632800*. (**F**) The ratio of maternal expression to paternal expression for each gene.

**Figure 2 ijms-21-09618-f002:**
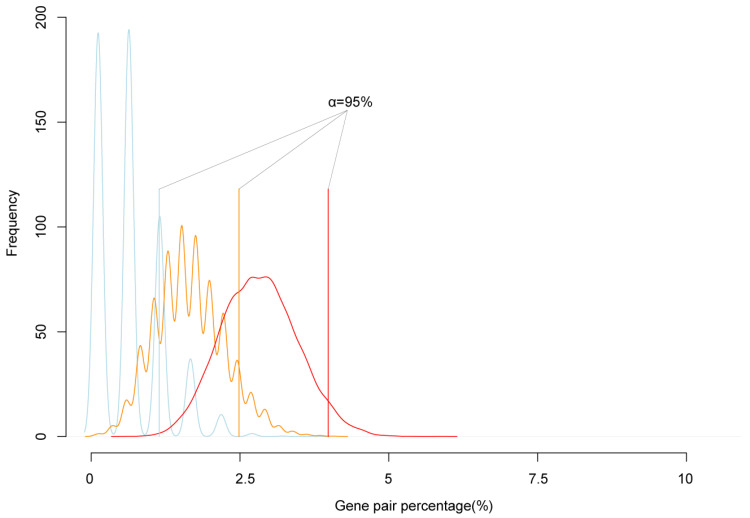
The simulation of the frequency of neighbor genes. The blue, orange, and red lines are the simulations of the percentage of neighbor gene accounted in randomly picked 200, 500, and 1000 genes from the genome, respectively. The solid blue, orange, red straight line is the 95% confidence interval of random neighbor genes percentage; the black straight solid line is the real neighbor genes in the detected imprinted genes.

**Figure 3 ijms-21-09618-f003:**
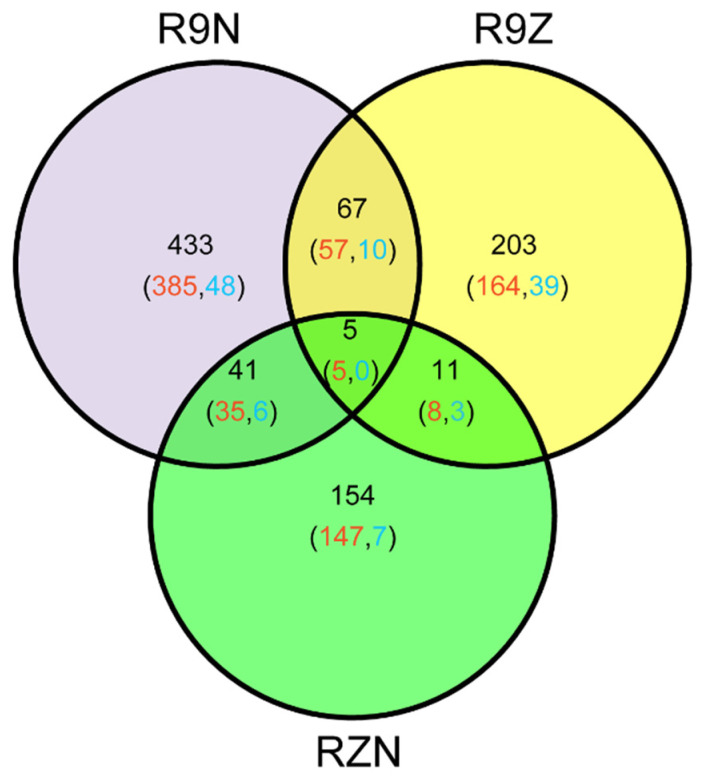
The Venn diagram of the genes identified in the three sets of reciprocal crosses. R9N, reciprocal hybrids of 9311 and Nip (9N + N9); R9Z, reciprocal crosses of ZS97 and 9311 (Z9 + 9Z); RZN, reciprocal hybrids of Nip and ZH11 (NZ + ZN). Red number represented the number of MEGs; blue number represented the number of PEGs.

**Figure 4 ijms-21-09618-f004:**
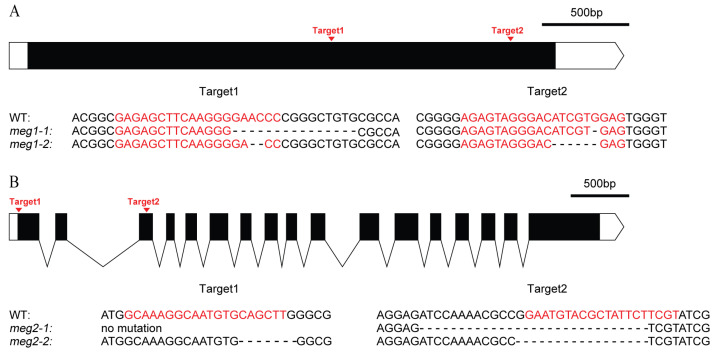
Genomic editing of two MEGs by CRISPR (Clustered Regularly-Interspaced Short Palindromic Repeats)/Cas9 (CRISPR associated protein 9). (**A**) Genomic editing and mutation types of *MEG1*. (**B**) Genomic editing and mutation types of *MEG2*. The black box and curve mean the exon and intron of gene, respectively. The white box and the white box with arrow mean 5′UTR and 3′UTR, respectively. The red letters are the targets of gene editing. “-” is the missing base.

**Figure 5 ijms-21-09618-f005:**
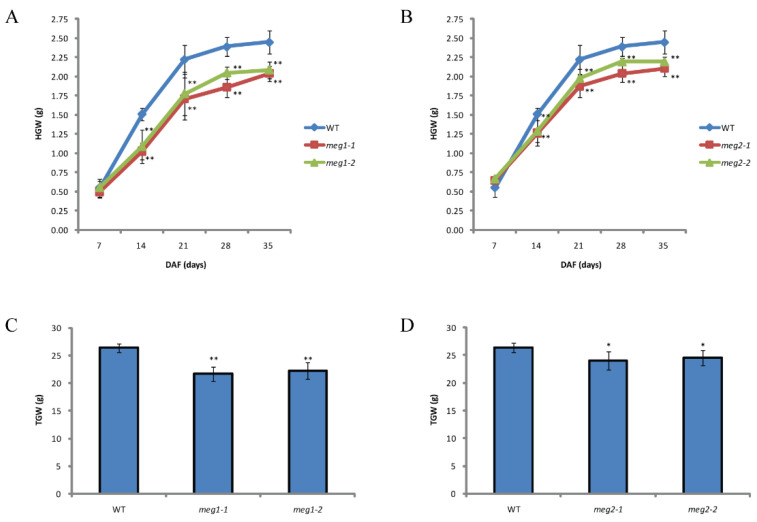
Comparison of grain filling between WT (wild type) and mutants. (**A**) Comparison of dynamic measurement of grain filling between wild type and two mutants of *MEG1*. (**B**) Comparison of dynamic measurement of grain filling between wild type and two mutants of *MEG**2*. (**C**) Comparison of thousand-grain weight (TGW) between wild type and two mutants of *MEG1*. (**D**) Comparison of TGW between wild type and two mutants of *MEG**2*. HGW, hundred-grain weight. Student’s t test was used for statistical analysis (*, 0.01 < *p* < 0.05 and **, *p* < 0.01).

**Figure 6 ijms-21-09618-f006:**
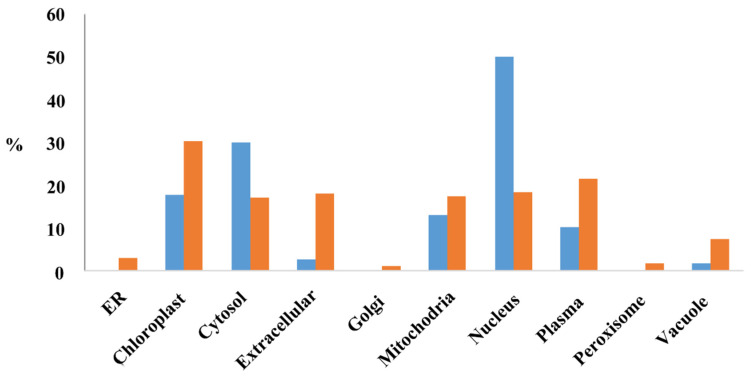
The subcellular localization of proteins encoded by MEGs and PEGs. Y-axis stands for the percentage of proteins located in each organelle. Blue, proteins encoded by PEGs; Orange, proteins encoded by MEGs.

**Figure 7 ijms-21-09618-f007:**
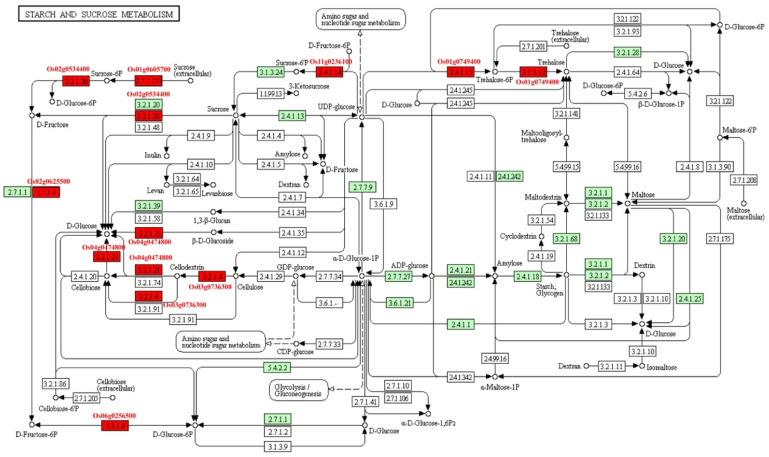
The pathway of starch and sucrose metabolism. Red box, MEGs involved in the pathway; Green box, the enzymes involved in starch and sucrose metabolism existed in rice; White box, the enzymes existed in other species.

**Table 1 ijms-21-09618-t001:** Single Nucleotide Polymorphisms (SNPs) and Insertion and Deletion (InDels) identified in each sample.

Sample Name	SNP Number	InDel Number	Expressed Genes ^1^	SNP Genes ^2^	InDel Genes ^3^
N9 endosperm	168,864	21,180	23,512	21,638	11,539
N9 embryo	202,828	26,279	23,092	22,850	9991
9N endospermrep1	67,783	6688	22,624	16,380	4370
9N embryo rep1	93,470	9588	24,265	18,022	5953
9Nendosperm rep2	56,332	4681	22,933	15,332	3089
9N embryo rep2	75,491	6478	24,801	16,792	4228
Total ^4^	229,065/189,303	34,809/28,026	26,346/24,558	24,598/24,558	12,499/10,864
9Z endosperm	36,319	2329	18,096	5484	799
9Zembryo	46,725	3210	19,484	5700	283
Z9 endosperm	29,136	1822	18,587	5220	681
Z9 embryo	52,875	3687	19,964	6151	1452
Total	59,941/41,470	4517/2825	25,528/25,021	8330/7180	1795/986
ZN endosperm	9279	341	23,573	2138	262
ZN embryo	14,735	523	24,477	2373	403
NZ endosperm	8842	279	23,320	1956	228
NZ embryo	12,758	491	23,464	2138	375
Total	16,776/11,870	689/421	25,256/24,866	2891/2462	492/307

^1^ The number of expressed genes detected in each sample. ^2^ The number of genes had SNPs between parents. ^3^ The number of genes had InDels between parents. ^4^ Total number of each statistical item of embryo and endosperm was listed in front and behind of slash.

**Table 2 ijms-21-09618-t002:** Summary of imprinted genes identified in three sets of reciprocal hybrids.

Hybrids	Endosperm			Embryo		
	Total ^1^	MEG	PEG	Total ^1^	MEG	PEG
R9N ^2^	546	482	64	0	0	0
R9Z ^3^	286	234	52	1	0	1
RZN ^4^	211	195	16	3	3	0

^1^ Total number of imprinted genes. ^2^ RN9 represented for reciprocal crosses of Nip and 9311. ^3^ R9Zrepresented for reciprocal crosses of 9311and ZS97. ^4^ RZN represented for reciprocal crosses of ZH11 and Nip.

**Table 3 ijms-21-09618-t003:** The detail information of 913 imprinted genes in each reciprocal crosses.

Gene Classification	R9N	R9Z	RZN
imprinted gene	546	286	211
Non-significant ^1^	260	21	23
no SNP ^2^	48	544	676
no ASR ^3^	60	62	4

^1^ The ratio between maternal ASRs and the paternal allele specific reads (ASRs) of each gene was not significantly deviated from the expected ratio of 2m:1p by Chi-square test. ^2^ There was no SNP between maternal genome and paternal genome. ^3^ There was SNP between parents, but there were no reads across the SNP.

**Table 4 ijms-21-09618-t004:** Five common MEGs detected among three reciprocal crosses.

Gene ID	Imprinting Type	Annotation
*Os01g0151700*	MEG	Similar to Short-chain dehydrogenase Tic32.
*Os07g0103100*	MEG	Similar to Glutamate receptor 3.4 precursor (Ligand-gated ion channel 3.4) (AtGLR4). Splice isoform 2.
*Os10g0340600*	MEG	Similar to Beta-galactosidase.
*Os11g0679700*	MEG	Glycerol-3-phosphate acyltransferase, Anther development, Pollen formation.
*Os12g0632800*	MEG	Protein kinase, catalytic domain containing protein.

**Table 5 ijms-21-09618-t005:** The detail information of three reciprocal crosses.

Cross Combination	Direct Cross	Reciprocal Cross
R9N ^1^	9311 × Nip ^2^	Nip × 9311
R9Z	9311 × ZS97	ZS97 × 9311
RZN	ZH11 × Nip	Nip × ZH11

^1^ R9N represents the crosses between 9311 and Nip including direct cross and reciprocal cross. ^2^ The female parent and the male parent are listed before and after “×” respectively.

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
