# Peer review of "Conserved Imprinted Genes between Intra-Subspecies and Inter-Subspecies Are Involved in Energy Metabolism and Seed Development in Rice"

_ijms, 2020, doi:10.3390/ijms21249618_

Round 1
Reviewer 1 Report
Review the manuscript ID: ijms-996644
This investigation identifies imprinted genes from embryo and endosperm in developmental seeds of rice one inter-subspecies and two intra-subspecies reciprocal hybrids.The authors used methods adequate to the tasks, the results obtained are correct and enrich the database of imprinted genes in cereals, expanding the understanding of genomic imprinting that is one of the most interesting and important events in the epigenetic regulation of gene expression.
The manuscript can be recommended for publication after certain corrections. It is necessary to give a list of abbreviations. It is desirable to shorten the Introduction, e. g. incomprehensible citing Shirzadi et al, 2011 (L. 68–76, can be deleted), and present the matters dealt with in Discussion more clearly; also to transfer the 2.9 “Important roles in seed development of conserved imprinted genes between species” from Results to the Discussion and make this section more interesting. The authors compare the imprinted genes in the endosperm of rice, maize, and Arabidopsis. It is advisable to know how the authors relate the differences in genomic imprinting of endosperm to the peculiarities of the development of endosperm in Arabidopsis seeds and cereal grains as well as the differences in accumulation of reserve nutrients (in the embryo cotyledons of Arabidopsis and in the endosperm of rice carypses).
The text contains a number of unfortunate expressions, e.g. “The plastid of endosperm that was similar to chloroplast was the organelle of starch synthesis” (L.317, 318) and many misprints. The text needs a remarkable improvement and the cited items a careful revision.
Reviewer 2 Report
The work argued if genomic imprinting could be influential in choosing the alleles coming from parents. Authors asked questions: weather imprinting is conserved between indica and japonica species and if interspecies hybrids could bring new imprinting patterns. Questions seems very legit seeing the differences between evolution of these two-rice subspecies. The rationale behind the study is very legitimate and authors try to address them using RNAseq data analysis supplemented by simulations.
Authors should represent the crosses from section 4.1 in graphical representation, understanding the same from text is bit complicated. This will make terminology like RN9, R9Z and RZN will be clearer for reader, rather mentioning them in table 2.
More details should be provided on methodology for simulation. Better explanations are needed how simulation could help in building a reliability for conservation of imprinting genes? For now, it is just another set of data.
Justification needed for choosing MEG1 and MEG2. Moreover, the knockout experiment need more proven validations. The grain filling measurement is not enough, specially the difference might be significant statistically but it is not striking enough to spoke for the conclusions made with these observations. The reverse genetic experiment would have been more appropriate by testing the gene both MEG and PEG classes.
Authors should extend their arguments in discussion. The objectives stated in introductions didn’t get well addressed in discussion and conclusion.
Some language editing is necessary to improve the grammatical mistakes and readability of MS. Specially the context where sentences are longer.
Line 189: genes were; line 466, line 185-188, line 271-273
Line 169: Figure cited wrong
Line 164: This expression was also…. Need more explanations
Line 236-238: Not clear what authors wish to communicate with “which was more than 21”?
Data availability and open access (GitHub) to claimed inhouse scripts should be made to ensure the reproducibility of the study.
